# State-of-the-Art Review of Ground Penetrating Radar (GPR) Applications for Railway Ballast Inspection

**DOI:** 10.3390/s22072450

**Published:** 2022-03-22

**Authors:** Shilei Wang, Guixian Liu, Guoqing Jing, Qiankuan Feng, Hengbai Liu, Yunlong Guo

**Affiliations:** 1Infrastructure Inspection Research Institute, China Academy of Railway Sciences Co., Ltd., Beijing 100081, China; wangshilei@rails.cn (S.W.); liuguixian@rails.cn (G.L.); fqk@rails.cn (Q.F.); liuhengbai@rails.cn (H.L.); 2School of Civil Engineering, Beijing Jiaotong University, Beijing 100044, China; gqjing@bjtu.edu.cn; 3Faculty of Civil Engineering and Geosciences, Delft University of Technology, 2628 CN Delft, The Netherlands

**Keywords:** ground penetrating radar, GPR, railway ballast, track inspection, ballast fouling, track geometry

## Abstract

In the past 20 years, many studies have been performed on ballast layer inspection and condition evaluation with ground penetrating radar (GPR). GPR is a non-destructive means that can reflect the ballast layer condition (fouling, moisture) by analysing the received signal variation. Even though GPR detection/inspection for ballast layers has become mature, some challenges still need to be stressed and solved, e.g., GPR indicator (for reflecting fouling level) development, quantitative evaluation for ballast fouling levels under diverse field conditions, rapid GPR inspection, and combining analysis of GPR results with other data (e.g., track stiffness, rail acceleration, etc.). Therefore, this paper summarised earlier studies on GPR application for ballast layer condition evaluation. How the GPR was used in the earlier studies was classified and discussed. In addition, how to correlate GPR results with ballast fouling level was also examined. Based on the summary, future developments can be seen, which is helpful for supplementing standards of ballast layer evaluation and maintenance.

## 1. Introduction

### 1.1. Problems with Ballast

Although slab tracks have developed rapidly in recent decades, ballasted tracks are still the main track type for all kinds of transportation (high-speed railways, heavy-haul railways, metros, etc.) [1,2]. In particular, ballasted tracks are also the main track type for general speed railways at speeds of 200–250 km/h in China [3,4].

The ballast layer is the main component of ballasted tracks, as shown in Figure 1, and is mostly made of crushed rocks [5,6]. It has the function of supporting the sleeper evenly [7,8], transmitting loading uniformly to the substructure (subgrade, bridge, and tunnel) [9,10], resisting the sleeper movements [11,12,13,14,15], and providing sufficient drainage [16,17,18]. Currently, all kinds of railways (high-speed railways, heavy-haul railways, intercity, and metros) are rapidly developing worldwide, such as with the Belt and Road Initiative. In addition, the existing railway lines need carful maintenance. For these two reasons, the ballast consumption is huge [19]. However, the suitable ballast resources (parent rock) are increasingly tight, and it is very difficult to find qualified ballast material for high-speed railways due to the high standards [20]. In addition, one challenge of applying ballasted tracks for higher speed trains is that fouling is rapidly generated in the ballast layer during service [21]. Therefore, developing a reasonable ballast layer maintenance plan, improving the inspection means, reducing ballast consumption, and controlling maintenance costs are the main development directions for ballast layers.

### 1.2. Problems Caused by Ballast Fouling

The fouling sources mainly include the following aspects [23]: (1) fines from ballast abrasion and small particles from ballast breakage, (2) infiltration from outside of ballast layer, and (3) degradation of other components.

(1) Fines from ballast abrasion and small particles from ballast breakage [24]. It was reported that up to 70% fouling is from ballast degradation for railway lines without weak subgrade or external intrusions such as sand and coal fines [25]. This percentage can differ for different transportation types. For example, in [5], the authors stated 20% (second largest percentage) is from ballast degradation in the coal transportation line. Nevertheless, ballast degradation is still the main fouling source.

Ballast particles undergo cyclic loading from trains [26,27]. The relative ballast movements produce fines, especially under high-speed train loading [28]. The loading energy is consumed mostly by the ballast particles, and a small part of the energy is transmitted to the subgrade [29,30]. The energy consumption means is the friction between ballast particles. The ballast particles bear high stresses, causing breakage (fracture, angularity loss, etc.) [31,32], especially under heavy-haul trains [33]. The ballast particles also deteriorate rapidly during tamping and stabilisation. Tamping breaks and wears ballast particles, while stabilisation wears ballast particles more than tamping [34].

(2) Infiltration from outside of ballast layer. The main infiltration into ballast layer is subgrade soil causing ballast pockets. The ballast pockets inevitably occur because under cyclic loading and ballast weight, the ballast and subgrade intrude into each other. According to [5], up to 58% of fouling comes from subgrade infiltration. The ballast pockets contain a mixture of ballast–soil–water and can be seen as highly fouled [35]. Another important infiltration is the material from freight transportation, for example, coal and sands [36,37]. In China, there is a railway line only for coal transportation, which needs ballast cleaning every year [38]. In addition, in desert areas, wind-blow sands are the main fouling source [39]. The sands increase the track stiffness and reduce the track resilience [40]. Finally, human excrement in most cases is ignored, but it is also an important fouling source. It can also contribute to vegetation growth on ballast layer surfaces [41].

(3) Degradation of other components. Other components in railway systems, such as the sleeper and catenary, also deteriorate and then drop into the ballast layer as fouling occurs. This fouling source does not occupy much of the fouling materials.

Fouling in the ballast layer usually causes the ballast layer to harden, ballast pockets, and mud pumping [42,43]. In particular, when mud pumping occurs, the ballast shear strength decreases significantly by 40% [44].

The ballast layer hardens when the fouling jams the voids between ballast particles; then, the stiffness of ballast layer increases, and resilience reduces [45]. Ballast pockets usually arise at the ballast–subgrade interface below the rail [22]. The high intensive stresses from the train are directly transmitted to the ballast and subgrade below the rail [9]. Then, the ballast particles and subgrade start to mix at the interface, producing ballast pockets [25]. Note that the soil from the subgrade is a crucial source of ballast fouling [5]. The ballast pocket usually stores a large amount of rainwater, causing mud pumping [46]. Mud pumping is the water together with fouling that is sucked up to the ballast layer surface, which normally occurs near the ballast pocket [35].

Due to fouling, the ballast layer condition gradually deteriorates [47,48], and the ballast layer loses its elasticity/resilience and drainage function. Ballast fouling is also a key indicator to assess the track condition, especially the track geometry. Poor track geometry affects smooth and safe rides [49]. More importantly, the track geometry must be maintained frequently if fouling is rapidly generated, for example, in freight railway lines for coal.

Ballast fouling is dealt with by ballast cleaning and ballast renewal [34]. Ballast cleaning sieves the fines in the ballast layer out, while ballast renewal replaces the old ballast layer with new ballast particles. Some large machinery and equipment are used for ballast layer cleaning, for example, sucking the coal fines at the surface of ballast by the vacuum. To enhance the fouled ballast layer and increase its shear strength, the most widely known application is the geogrid. Multiple studies have been performed on this topic; see [50,51,52,53,54,55,56].

In summary, we reviewed the ballast layer function, fouling source, fouling consequences (effects on ballast layer function), fouling cleaning and enhancement methods. However, ballast fouling inspection is still an unsolved problem. The fouling is mostly generated and accumulates at the ballast–subgrade interface and then gradually accumulates until reaching the ballast layer surface. The fouling level (or fouling rate) is very hard to observe from the ballast layer surface or to judge from the in-train dynamic responses until mud pumping and serious ride comfort occur.

The maintenance for ballast fouling is only based on the gross passing load, which is a rough method for ballast layer condition prediction and ballast fouling content estimation [34,57]. This results from the fact that the development of inspection technology and equipment cannot meet the rapidly increasing train speed and freight weight. The ballast fouling sources vary, as mentioned above. Therefore, the fouling level corresponds to many influencing factors, such as the gross passing load, transportation type, ballast material, special structures, region, and climatic conditions [58]. Therefore, predicting the fouling level only based on gross is unreasonable and sometimes provides incorrect maintenance decisions (e.g., over maintenance, lack of maintenance). In addition, the fouling index of the ballast layer is obtained by sieving the ballast samples drilled from the field [59]. This method requires considerable manpower and resources, which is a very limited method and is rarely applied in normal railway inspection.

### 1.3. Examples of GPR Applied to Ballast Layer

Addressing unreasonable ballast cleaning decisions, GPR has been widely studied for its capability and application to ballast fouling inspection. This paper summarised the GPR application for ballast layer inspection. We aimed to improve the inspection means and further develop a reasonable ballast layer maintenance plan, by which the ballast consumption and maintenance costs can be significantly reduced.

In the past 20 years, researchers worldwide have carried out many in-depth studies on ballast layer inspection along with the improvement in railway inspection technology and equipment [60,61,62,63]. Some of the technologies have been applied widely in developed countries, such as Europe and the United States. Additionally, they have established a preliminary quantitative evaluation standard (using GPR) for ballast layers. The evaluation standard is mostly based on the American railway and fouling index, which may be more applicable for the ballast layer in the USA. For instance, to verify the applicability of the system on US railways, an approximately 18 km railway line in Wyoming was selected for testing. Thirty-two locations were selected for drilling and testing fouled ballast samples. The relationship between the Selig fouling index [25] of the samples and the ballast fouling index from GPR inspection are shown in Figure 2, with a linear correlation coefficient of 0.87.

### 1.4. Examples of GPR Application for Track Geometry

Defects in track geometry are one of the most important factors affecting ride comfort and safety [64]. Studying the relationship between the track substructure and track geometry (deterioration rate) is important for track maintenance and track geometry inspection [65,66,67,68]. For ballasted tracks, track geometry defects are in most cases related to the ballast layer conditions [69]. Therefore, the correlation between ballast layer condition and track geometry provides an accurate evaluation for the track condition.

At present, all types of studies on ballast layer conditions and track geometry defects focus on the relationship between ballast layer deformation and track geometry irregularity [65,66,70,71]; in particular, focus has been placed on the deterioration of the track quality index in railway sections with serious ballast layer defects (e.g., mud pumping) [53,72,73,74]. Studies have shown that ballast layer deformation and track irregularity have an approximate proportional relationship. In addition, the influence of ballast layer deformation on track irregularity increases with time [75,76,77,78]. Tests on several 200 m intervals have shown a linear deterioration in track geometry between two maintenance periods [79,80,81]. Linear prediction models for track geometry degradation have been widely proposed to predict the development of the track quality index (TQI) [82,83,84,85].

Limited studies have been found on correlating ballast fouling levels with track geometry defects (deterioration rates). In the study by [86], track geometry defects frequently occurred at two locations. Then, GPR detection was used to see the ballast layer condition, and field drilling was carried out at the two locations. It was found that both locations exhibited significant ballast fouling and abnormally high water content [86,87,88,89].

The relationship between ballast layer performance and track geometry defects was discussed in [90,91,92]. This was the first attempt to quantify the relationship between the probability of track geometry defects and the ballast fouling level. It is also noted that few studies have predicted the relationship between the development of track geometry defects and ballast fouling level. In particular, there is a lack of quantitative analysis of the correlation between the ballast fouling level and the development of track geometry defects. Therefore, there is no accepted, empirically validated theory of the relationship between the development of geometric defects and the ballast fouling level.

In [93,94], the GPR results were used for track modulus estimation, mainly on the ballast layer and sub-ballast layer. In addition, some features of different ballast and sub-ballast layers were compared. For example, some of these railway lines applied geocells and asphalt at the interface of the ballast layer and sub-ballast layer. The results show that the GPR data can estimate the track modulus with an accuracy of ±3.4 MPa, as shown in Figure 3.

**Motivations.** The gross passing load, transportation type, ballast material, special structures, region, and climatic conditions are all different between each country. Therefore, this paper summarises earlier studies on GPR applications, which were performed in each country under different field conditions. This is very meaningful for supplementing standards for ballast layer inspection.

## 2. Electromagnetic Properties

### 2.1. GPR Detection Principle

The ballast layer is made of hard crushed rocks with a certain particle size distribution (also named grading/gradation). A typical ballasted track with a ballast layer is shown in Figure 1. The porosity of the clean compacted ballast layer is in the range of 30–40%, which needs to guarantee that (1) the train load is transferred evenly to the subgrade, (2) sufficient drainage is provided, and (3) the track elasticity is maintained. The ballast layer thickness is generally in the range of 250–350 mm (from sleeper to layer bottom), while the total thickness (from layer surface to the layer bottom) is generally in the range of 470–640 mm.

The principle of GPR detection technology is shown in Figure 4. The transmitting antenna emits electromagnetic waves to the inside of ballast layer. When the waves encounter a boundary layer or area with different dielectric properties, the waves are reflected and scattered. Then, the receiver antenna records the return wave time and amplitude to form a single waveform, as shown in Figure 4. The single waveforms are obtained along the railway line, which can be gathered to form a two-dimensional radar image. By analysing the single waveform signal and the 2D radar image, indicators that can reflect ballast layer thickness, ballast fouling level, and drainage are obtained.

### 2.2. Ballast Layer Electromagnetic Properties

#### 2.2.1. Factors Influencing Electromagnetic Wave Propagation

The propagation of electromagnetic waves in a medium follows Maxwell’s equations. In most cases, ballast is nonmagnetic. Then, the propagation of electromagnetic waves can be described by the following two equations with time as the variable.
(1)$$\nabla E=-\mu \frac{\partial H}{\partial t}$$
$$\nabla H=\varepsilon \frac{\partial E}{\partial t}+\sigma E$$
where *E* is the electric field strength, *H* is the magnetic field strength, *ε* is the dielectric constant, *μ* is the magnetic permeability, and *σ* is the electrical conductivity. *ε*, *μ*, and *σ* are the three parameters that determine the propagation of electromagnetic waves in the ballast layer.

The dielectric constant *ε* characterises the ability of a medium to be polarized by an electromagnetic field. The dielectric constant determines the speed of electromagnetic wave propagation (*v*) in the medium, as shown in Equation (2):(2)$$v=\frac{c}{\sqrt{\varepsilon }}$$
where *c* is the propagation speed of electromagnetic waves in a vacuum (i.e., the speed of light). The dielectric constant of air is approximately 1, which results in an EM wave velocity of approximately 0.299792 m/ns. The dielectric constants of common materials in ballast layers are shown in Table 1. The dielectric constant of water is much larger than that of the other materials, and its dielectric constant is related to the frequency of the electromagnetic wave. Because the magnetization ability of water in a low-frequency electromagnetic field is strong, the dielectric constant is high. While the phenomenon of relax occurs for water molecules in an electromagnetic field, the water magnetization ability is weak. Then, the water dielectric constant is low. Nevertheless, research shows that Relax is not apparent at natural ambient temperatures with an electromagnetic wave frequency of 2 GHz. The 2 GHz frequency is the most popular frequency in recently published papers [23,95,96].

The ballast layer is a mixture of ballast particles and the fouling contained within ballast voids. The dielectric constant of a mixed medium (consisting of two or more anisotropic materials) is usually calculated using the complex refractive index method (CRIM) [97], as shown in Equation (3).
(3)$$\varepsilon_{b}={\left(\sum \rho_{i}\varepsilon_{i}^{\alpha }\right)}^{1/\alpha }$$
where *ε_b_* is the dielectric constant of the mixture, *ρ_i_* is the volume percentage of each material of the mixture, *ε_i_* is the dielectric constant of each component, and *α* is a parameter that depends on the spatial structure and the angle between the mixture and the electric field; generally, *α* is taken to be 0.5.

For example, there is a ballast layer with a porosity of 35%. The dielectric constant of the fouled ballast layer can be calculated according to the CRIM (Equation (3)). The fouled ballast layer can be seen as one medium made by ballast, air, water, and fine particles. Considering water and fine particles filling ballast voids from 0% to 100%, the mixture dielectric constants are calculated as shown in Figure 5. When all the ballast voids were filled by water, the mixture dielectric constants increased from 4 to 22. When all the ballast voids are filled by fine particles, the mixture dielectric constants increase by only 3. This means that water is the determining factor for the dielectric constant of the ballast layer.

In addition, Figure 5 presents the results under ideal conditions, which assumes that fine particles and water can exist independently in ballast voids. However, water mainly relies on fouling. Studies [98,99,100,101] show that the normal fouling material (mostly soil) in the ballast layer can store less than 30% water to the total fouling volume. The small ballast particles can contain less than 20% water, and the loose fine particles of coal can hold less than 10% water. Therefore, the dielectric constants of the fouled ballast layer are usually distributed in the purple area in Figure 5. Generally, the clean ballast layer dielectric constant is approximately 4, while the dielectric constants of the top-fouled ballast layer (all voids are filled by fouling) range from 7 to 11, which depends on the water holding capacity of the fouling.

Magnetic permeability (*μ* in Equation (1)) affects the speed of electromagnetic wave propagation and energy attenuation in a medium. The magnetic permeability of ballast and common fouling materials is generally low, for which this is usually not considered in earlier studies.

Electrical conductivity (*σ* in Equation (1)) has a negligible effect on the speed of electromagnetic wave propagation when considering dry lithic material, but it mainly triggers energy attenuation in signal transmission. The electrical conductivity is only dependent on the electromagnetic wave frequency for some materials, such as water, but in the range of the GPR signal band (MHz–GHz), the electrical conductivity influence on the results can also be ignored [102,103].

#### 2.2.2. Signal Reflection

The direction of electromagnetic waves is approximately perpendicular to the horizontal plane during GPR detection of the ballast layer. The perpendicularly incident electromagnetic waves are reflected at the interface of adjacent nonmagnetic materials (e.g., ballast, air). The reflection magnitude is determined by the reflection coefficient of the interface, which is calculated by Equation (4).
(4)$$r_{j,j+1}=\frac{\sqrt{\varepsilon_{j}}-\sqrt{\varepsilon_{j+1}}}{\sqrt{\varepsilon_{j}}+\sqrt{\varepsilon_{j+1}}}$$
where *r_j_*_,*j*+1_ is the reflection coefficient, *ε_j_* is the dielectric constant of the material located at the upper part of the interface, and *ε_j_*_+1_ is the dielectric constant of the material located at the lower part of the interface.

Equation (4) shows that the larger the dielectric constant difference between the lower part of the interface and the upper part of the interface, the greater the absolute value of the reflection coefficient. This indicates that the receiver antenna receives a stronger reflection signal. Because the air dielectric constant is close to 1 and the dielectric constant of ballast layer is 4–11, the first clear reflection occurs at the air–ballast interface, as shown in Figure 6.

Figure 6 demonstrates that the reflection at the air–ballast interface increases with increasing ballast fouling level. Because a high fouling level has a high dielectric constant, the reflected signal becomes stronger. In China’s normal railway lines, the subgrade is made of compacted soil, and sub-ballast (small crushed rocks) is used to cover the subgrade surface [104]. They both have high density, leading to their high dielectric constant at 9–14, which is larger than that of the normal ballast layer. Therefore, another distinct reflection can be found at the ballast–subgrade interface (Figure 6). Distinct interface reflection can also occur inside a ballast layer when the ballast layer is severely fouling and made of double layers (one clean ballast layer and one fouled ballast layer). Two other reasons causing clear reflection are the high water content of the ballast layer and mud pumping. However, when all the ballast layer voids are filled by fouling, the dielectric constants between ballast layer and subgrade are similar, resulting in a weakened or missing reflection in the bottom, as shown in Figure 6. At the positions that lack reflection, we dug the ballast layer to observe the fouling degree, which showed that the ballast layer is heavily fouled by soil and coals.

#### 2.2.3. Signal Scattering

When the electromagnetic wave encounters an object with a size close to the wavelength and the object’s dielectric properties have a large difference from the surrounding medium, the scattering response of the electromagnetic wave is strong. The scattering response level is related to the ratio *α* of the scatter size to the electromagnetic wave wavelength, as shown in Equation (5), where *D* is the diameter of the scatter and *λ* is the wavelength of the electromagnetic wave. When *α* ≪ 1, it is named Rayleigh scattering with a low scattering response, and when *α* ≈ 1, it is named Mie scattering with a strong scattering response.
(5)$$\alpha =\frac{2\pi D}{\lambda }$$

Equation (5) shows that a higher frequency and shorter wavelength lead to a larger *α* value and stronger scattering response. Geophysical Survey Systems, Inc. (GSSI, in New Hampshire, United States) discovered that the voids in ballast layer can be considered as the scatter with 2 GHz high-frequency antenna, which means scattering response occurs around the void in ballast layer during propagation. The void size is related to the particle size distribution, ballast layer compaction, and fouling level. Studies have reported that the void size is generally in the range of 5–15 mm. Barrett [86] calculated the scattering efficiency (within the usual GPR frequencies) of different void sizes (1, 2, 5, 10, 20, and 50 mm) in the ballast layer, as shown in Figure 7. The scattering efficiency nonlinearly increases with increasing electromagnetic wave frequency. In the 2 GHz frequency range, the scattering efficiency is approximately 100,000 times higher for void sizes in the range of 5–20 mm than for void sizes in the range of 1–2 mm. The scattering response of voids in the ballast layer in the 2 GHz frequency range is strong, which can also be observed in Figure 8. When the voids in the ballast layer are filled by fouling, the electromagnetic wave scattering response decreases, providing the possibility of evaluating the fouling level through analysis of the scattered signal within the ballast layer.

## 3. GPR Applications

To optimise the track maintenance schedule and scientifically develop ballast layer maintenance strategies, many studies have been carried out on ballast layer inspection with GPR. Along with signal analysis methodologies and radar equipment improvement, these studies have proposed a series of ballast layer identification methods based on laboratory tests, field measurements, field tests, etc. In general, these identification methods can be classified into two aspects: qualitative analysis methods based on radar image processing and quantitative analysis methods based on radar signal information. Quantitative ballast layer condition identification has received widespread attention because it can carry out rapid railway network condition monitoring and management. This section focuses on the quantitative assessment of ballast layer conditions and compares the research and application status of related technologies in the past 10 years. These methods can be divided into three categories: the dielectric constant method, time-domain and frequency-domain analysis method, and high-frequency signal scattering analysis method.

### 3.1. Dielectric Constant Method

Benedetto et al. [105] applied a resin open container filled with ballast particles following European standard particle size distribution (PSD) [20] and pulverized soil to simulate four different ballast fouling levels. The test setup can be seen in Figure 9. The 1 GHz and 2 GHz air-coupled antennas used in their study are made by IDS company, Italy. Because the ballast layer has already been measured as 0.55 m, the dielectric constants of ballast layers with different fouling levels were calculated by using the time-domain signal extraction and ballast layer surface reflection. The calculated dielectric constants were also compared with the dielectric constants calculated by CRIM (Equation (3)). The results show that the dielectric constants are in the range of 3.51–5.35. The dielectric constant increases with the fouling level. However, this method is not suitable for the evaluation of the dielectric properties of ballast layers in the field, even though the obtained constants match well with the constants calculated by CRIM. This is on the condition that the ballast layer thickness has already been known. Then, by time-domain signal extraction, the dielectric constant is easy to obtain.

Artagan et al. [63] in an indoor laboratory built a ballast box filled by ballast samples with different fouling sources and water contents, as shown in Figure 10. The 2 GHz air-coupled and 400/900 MHz dual-band ground-coupled antennas are used. The ballast layer thickness is measured, which is used to calculate the dielectric constants of the ballast samples. A linear regression model between the dielectric constant and the fouling level considering different water contents was established. In addition, the GPR signals of the 138 m field railway line were obtained, and in the line, 9 holes at different locations were drilled to obtain fouled ballast samples (analyse the field fouling level by sieving). The results show that the dielectric constants of the field railway line with different fouling levels and the constants measured by laboratory test results can have a good match. This means that the dielectric constant method can identify the fouling level and water content of the ballast layer.

Discussion. By establishing a regression model between the dielectric constant and the ballast fouling level, it is theoretically feasible to estimate the fouling level of the ballast layer based on the measured dielectric constant of the field railway line [106]. However, the analysis results in Figure 5 show that the dielectric constant of ballast layer is an indicator that is multicorrelated with the types of fouling, the fouling level, and water content. This means that the same dielectric constant has multiple possibilities that lead to this result [107]. The actual thickness of the ballast layer is not known in the field, and the ballast–subgrade interface needs to be determined when using the dielectric constant method. There are currently no fast and mature detection methods to determine the type of fouling, water content, and actual thickness of the ballast layer, which still need to be determined by in situ drilling and indoor sieving. Therefore, this method still needs more development in these directions until it is applied in field ballast layer inspection.

### 3.2. Time Domain and Frequency Domain Analysis Method

Silvast et al. [108] proposed a method to quantify the fouling level in the ballast layer by using spectral domain integration to construct indicators. The fouling accumulates in the voids of ballast layer, which causes the dielectric increase. Then, the high-frequency GPR signal shows a stronger attenuation. The step-frequency 3D geological radar system produced by the 3D-Radar company was used to acquire the electromagnetic waves of the 173 km long railway line in Finland. The single channel waveform was Fourier transformed, and the peak values were normalised. The area of the frequency-domain signal in the range 0–1500 MHz was obtained by integration, as shown in Figure 11a. A length of 5 m was used as the base unit to average the frequency-domain signal area to obtain the GPR fouling index, and more than 60 locations were selected for drilling in situ fouled ballast samples. The drilling locations were in the area of 30–40 cm below the upper surface of the sleeper and at the ballast crib, which was 20 cm wide. The samples were sieved, and the fouling index (named the ballast degradation number) was calculated according to Equation (6). The linear correlation coefficient between the GPR fouling index and the ballast degradation number was found to be 0.87, as shown in Figure 11b. It is also proposed that ballast layers should be classified into clean, fouled, and highly fouled categories according to the GPR fouling index. It is suggested that the ballast fouling level of railway lines can be evaluated every 1 km length, which helps effectively decide the sections for ballast cleaning.
(6)$$FI_{D}=P_{1}+P_{8}+P_{25}$$
where *FI_D_* is the fouling index and *P*_1_, *P*_8_, and *P*_25_ are the percentages of the overall mass of ballast passing through 1, 8, and 25 mm sieves, respectively.

De Bold et al. [109] built a 10.7 m long full-scale model of ballast layer, set up sections with different degrees of soiling, sampled at the core of the road between each pillow, as shown in Figure 12a. In this study, the fouling index proposed by Ionescu [110] is used; see Equation (7). Ground-coupled antennas (500, 900, 1600, and 2600 MHz) were used to collect GPR signals. Scatter analyses of the GPR waveforms were proposed, including featuring area, axis crossing, and inflexion point, as shown in Figure 12b. The results show that the linear correlation between the scanned area of the 500 MHz antenna full time range and the fouling index is high, and the correlation coefficient is 0.92 when the two antennas are in the parallel direction (see Figure 12c).
(7)$$FI_{I}=P_{0.075}+P_{14}$$
where *FI_I_* is the fouling index calculated using the Ionescu method, and *P*_0.075_ and *P*_14_ are the percentages of the overall mass of ballast passing through 0.75 and 14 mm sieves, respectively.

Shangguan et al. [111] proposed a quantitative evaluation method for the ballast fouling level based on wavelet transform. Specifically, four ballast layer samples at different fouling levels were built indoors, and each sample size was 1.5 × 1.5 × 1.2 m. The fouling material was chosen as dry soil. The GPR signal waveform is decomposed into a series of wavelets. The direct coupled wave and surface reflection wave are removed. The specific and effective wavelets are combined and coupled, and their standard deviations are calculated. The fouling indices (proposed by Selig [25]) of each sample were calculated based on sieving (see Equation (8)). A quadratic function is taken to fit the standard deviation of the GPR signal to the Selig fouling index, and then the correlation between the GPR signal and fouling level is established. Subsequently, a vehicle equipped with a GPR system (suitable for both railways and pavement) was used to validate the results by inspecting a 25 km railway line when the weather was no rain. Six locations along this railway line were chosen to drill fouled ballast samples. The correlation coefficient of the GPR fouling index and the Selig fouling index can reach 0.95, which is a very high value.

Bianchini et al. [112] used a 30 m long section of railway line in Italy as a test section. The section was divided into 10 parts. The crib ballast was replaced by different fouled ballast with two types of fouling, i.e., small ballast particles and soil. Different fouling levels and water contents in the fouling were considered. The 1 GHz and 2 GHz air-coupled antennas were used to obtain the GPR signals in three directions. The air-ballast signal reflection amplitude and signal frequency-domain characteristics were compared and studied (see Figure 13). The results show that the standard deviation of the air-ballast signal reflection amplitude in the transverse direction of the 1 GHz antenna was found to be a 3-way function of the Selig fouling index. There is a clear reduction in the frequency-domain signal amplitude and integrated area of the 2 GHz antenna on the fouled ballast layer at low humidity levels compared to the clean ballast.
(8)$$FI_{S}=P_{0.075}+P_{4.75}$$
where *FI_S_* is the Selig fouling index, and *P*_0.075_ and *P*_4.75_ are the percentages of the overall mass of ballast passing through 0.075 and 4.75 mm sieves, respectively.

Earlier studies based on GPR carried out quantitative evaluation of the ballast fouling level. They proposed the use of spectral domain integration area (feature area), scanning area, wavelet signal standard deviation, air-ballast surface signal reflection amplitude standard deviation, and other time-frequency indicators such as the fouling indices. All these studies established the relationship between the GPR fouling indices (indicators) and the fouling indices obtained by in situ sample sieving. A linear relationship was found between the GPR indices and fouling indices based on in situ sample sieving when using the low-frequency GPR signal (approximately 500 MHz). The relationship between the GPR indices (based on the standard deviation of the time-domain GPR signal) and fouling indices based on in situ sample sieving at the medium- and high-frequency antennas (1000–2000 MHz) tends to be complex as a high regression phenomenon. The reason may be the limited number of samples. Another direction is that the GPR signals were collected mostly in the dry field or in the laboratory, for which the time domain and frequency domain of the GPR fouling index in the humid field ballast layer have not yet been correlated with the fouling index by sieving. To validate the GPR application, the relationship between the GPR fouling index and fouling index by sieving needs to be established in conjunction with the PSD. Differences in calculating ballast degradation or fouling were found in different countries due to different PSD standards.

### 3.3. High-Frequency Signal Scattering Analysis Method

In 2005, Roberts et al. [59] compared the ability of 1 GHz and 2 GHz antennas to reflect the ballast fouling level at TTCI (Transportation Technology Center, Inc., Pueblo, USA) in the USA. They found that the scattering response of electromagnetic waves (2 GHz antenna) occurs strongly at voids in the ballast layer [113]. Preliminary radar data processing ideas were proposed for 2 GHz antennas. Subsequently, the US Department of Transportation’s Federal Railroad Administration set up the Geophysical Radar Track Underline Evaluation Research Project, undertaken by GSSI and the University of Illinois [114]. To investigate the applicability of high-frequency GPR in ballast layer condition inspection, the research team considered the effects of different factors, such as the cleaning history, tamping, subgrade (sandy, soil, or expansive soil), and climatic environment (dry, rainy). They carried out GPR inspection on a 614 km long railway line. The GPR system consisted of three 2 GHz air-coupled antennas (including three transmitters and three receivers) and a Sir20 mainframe on a vehicle (which can be used for both rail and road), as shown in Figure 14a. The vehicle inspection speed was up to 40 km/h [23]. During the inspection, the antenna (bottom) was 350–410 mm upon the ballast layer surface, and the inner side antenna was 600 mm away from the sleeper end to reduce the rail influence on the signal. They also developed different colours to be used to present different ballast fouling levels based on the scattering response amplitude (see Figure 14b). The radar signal data acquisition and processing means are shown in Figure 14c. The difference is that the traditional radar image processing method applies background removal and gain adjustment [103], while Hilbert transform and smoothing of the transformed signal were used. Forty locations were selected for in situ drilling fouled ballast samples. The ballast samples were sieved, and moisture testing was performed. The results showed that the thickness of the clean ballast layer (calculated based on the GPR colour image) matched well with the drilling thickness result.

Based on Roberts’ research, Sadeghi et al. [45] proposed a method for classifying ballast fouling levels using a weighted count of the different colour areas to classify ballast fouling. GPR inspection was performed on a 17 km passenger and freight mixed railway in Iran. The ballast fouling index was built based on the results of each railway line length at 200 m. The results were proposed for use in tamping and ballast cleaning strategies.

Based on the scattering response within the ballast layer with high-frequency GPR signals, ZeticaRail developed a ballast layer inspection system, which focuses on the evaluation of ballast layer thickness and fouling. Two indices were built: the ballast depth exceedance (BDE) and ballast fouling index (BFI) [89]. The BFI is calculated from the high-frequency signal scattering response. The BFI must be calibrated using in situ sampling sieving results. This achieved uniformity between the BFI values and the Selig fouling index (commonly used by US railways). Based on the Selig fouling index, five fouling levels were classified as clean, medium clean, medium fouled, fouled, and highly fouled. For the purpose of bed condition management, the test system integrates the radar image, ballast thickness, ballast fouling classification, and 2D colour fouling image into the output.

ZeticaRail also applied the system to a heavy haul line for coal transportation from the Hunter Valley to northern Sydney, Australia [87]. Multi-aspect information on the ballast layer is shown in Figure 15. It was more appropriate to calibrate the BFI values by using the percent void contamination (PVC) index for ballast fouling on coal lines than by the Selig fouling index. The PVC is calculated using Equation (9) [115].
(9)$$\mathrm{PVC}=\frac{V_{2}}{V_{1}}$$
where *V*_1_ is the volume of voids in the clean ballast layer and *V*_2_ is the volume of particles less than 9.5 mm filling in the voids in the ballast layer.

Discussion. The study shows that the thickness of the clean ballast layer can be obtained based on 2 GHz GRP high-frequency scattering response analysis. In situ sieving results can be used to calibrate the GPR fouling index. This helps achieve precise management for the fouled ballast layer. Moreover, the thickness of the clean ballast layer can be used to assess the drainage capacity and reflect the fouling level [116]. The ballast fouling level can be used to guide ballast cleaning and renewal planning and strategies. Although the high-frequency scattering signal response analysis method has been validated in many studies, research into the influence of moisture on this method is still not known. Addressing the issue of moisture identification, multi-offset GPR data are introduced. By collecting GPR data in the multi-offset common midpoint geometry, the radar profile is improved, and it also allows for an interpretation of subsurface variation in water content [117,118]. The soil moisture is identified in agriculture by the application of GPR. Non-destructive determination of soil water content was found to be possible with GPR in practically continuous and detailed sections [119]. The researchers also analysed the common surface reflection and full-wave inversion methods to retrieve the soil surface dielectric permittivity and correlated water content from air-launched ground-penetrating radar (GPR) measurements [120]. A later study showed that the full-waveform GPR allowed accurate estimation of the near-surface water content; thus, it was able to monitor the evaporation process [121]. Some researchers may argue that the high moisture may come from the high ballast fouling, but this is not always the case. In the ballast layer, the difference in dielectric properties between the clean ballast layer and fouled ballast layer with high water content will have a clear interface. However, for the low water content ballast layer, knowing the boundary between the clean ballast layer and fouled ballast layer remains a problem. In addition, the influence of moisture in the ballast layer on the scattering response amplitude and the applicability of the GPR fouling index (established in a dry environment) to complex field conditions are also worthy of attention. Most importantly, to evaluate the ballast fouling level, it is necessary to establish a specific calibration between the GPR signal and the fouling index considering the different types of fouling (e.g., dedicated coal transport lines where the fouling source is mainly fine-grained coal).

### 3.4. GPR Application

Current high-frequency GPR inspection equipment has been developed and applied on a large scale in several countries in Europe and the USA. The UK uses GPR to carry out inspections at least once a year on highly used railways. Ireland has applied GPR inspection to the country’s longest Dublin–Cork railway line. They integrated the GPR inspection results into their track quality management system together with the track geometry quality index [87]. The US has applied GPR inspection to over 5000 km of Midwest coal lines for guiding ballast cleaning and track geometry defect analysis. The TTCI led the studies of GPR application on field railway lines, such as comparing six different railway lines on ballast layer conditions, high-frequency GPR carriers and application on multiple field conditions, and high-frequency GPR application on ballast fouling evaluation. These studies were carried out on the TTCI heavy-haul loop line and other typical railway mainline sections between 2015 and 2019 [90,91,92].

The GPR system was integrated into the DOTX-220 rail inspection vehicle of the US Federal Railroad Administration in 2020, realising the simultaneous detection of track geometry and ballast layer conditions. The vehicle integrates a medium-frequency and high-frequency GPR and laser scanning unit to provide rapid quantitative inspection of the ballast layer fouling level and ballast layer thickness, track profile, subgrade physical damage and subgrade slope performance. These inspections will support the plan of ballast cleaning and renewal, allocation of ballast resources, assessment of the ballast cleaning quality and knowing railway line surroundings. The functions of the integrated equipment are shown in Table 2.

To explore the ballast layer cleaning and renewal standards suitable for a coal-hauled railway line in China, a high-frequency GPR system from the UK Zetica company was used in 2017. The system was installed on the integrated inspection vehicle with a normal detection speed of 80 km/h. The equipment setup is shown in Figure 16. Based on the in situ drilling test and fouled sample sieving results, they obtained the fouling index of the specific locations, which were rapidly fouled. The fouling index by sieving was calculated by the mass percentage of particle sizes less than 25 mm. The GPR fouling index was calculated through the UK Zetica GPR system. The two fouling indices were correlated to determine the ballast fouling level standards; see Table 3. The system can also detect the thickness of the ballast for evaluating the quality of ballast cleaning; see Figure 17.

### 3.5. Ballast Layer Evaluation Incorporating GPR Inspection and Track Geometry

Ireland leads the studies on the application of ballast layer inspection and condition evaluation. An indicator system has been developed for ballast layer and track quality based on track inspection and GPR. The aim is to establish an accurate managing system for the ballast layer by classifying the ballast layer conditions. The indicator system consists of condition evaluations for both ballast layer and track.

#### 3.5.1. Ballast Layer Indicators

The main indicators for the evaluation of ballast layer are given as follows.
(1)The ballast depth exceedance (BDE) represents the difference between the actual ballast layer thickness and the required ballast layer thickness.(2)The layer roughness index (LRI) indicates the dispersion of the ballast layer thickness over a given length (5 m and 20 m). The LRI is used to highlight areas of rapid variation in ballast layer thickness, which may be associated with subgrade failure or mud pumping.(3)The trackbed drainage quality index (TDQI) indicates the drainage adequacy of the ballast layer. It measures the relative humidity levels at the ballast–sub-ballast interface at both ballast shoulders from a depth of 5 feet (approximately 1.5 m).

#### 3.5.2. Combined Track Quality Index

The combined track quality index (CTQI) is a combination of three GPR indicators: ballast depth exceedance (BDE), 20 m layer roughness index (LRI), and trackbed drainage quality index (TDQI). The calculation equation is shown as follows.
(10)CTQI=LRI+2×BDE+TDQI

Using the CTQI values, the ballast layer conditions were classified into four levels, namely, Levels 1, 2, 3, and 4. The severity of the ballast layer condition increases with a smaller number of CTQI levels. In addition, the track geometry quality index (QI) was also classified into 4 levels (Levels 1, 2, 3, and 4), with a smaller number indicating a more serious defect of track geometry [87]. QI2 track quality index.

The QI2 index combines the combined trackbed quality index (CTQI) and the track geometry quality index (QI) to reveal, as one indicator, the railway line sections of frequent track geometry deterioration that may or may not be related to ballast layer conditions and the railway line sections where poor ballast layer conditions can easily lead to subsequent rapid track geometry deterioration. The QI2 index is designed to ensure weighting when the track geometry QI is at a severe level. The other levels of QI and CTQI are classified by a rule matrix combining the problem severity into four levels, with different colours (i.e., black, red, yellow, and green), indicating very bad, bad, moderate, and good track conditions, respectively. The current rule matrix relating the CQTI and track inspection QI indices was obtained based on GPR detection. The rule matrix is shown in Figure 18.

#### 3.5.3. Form of Results

The integrated presentation of each indicator of the track and ballast layer is shown in Figure 19. CTQI and QI2 are presented in strips by the unit length at 5 m, and an integrated interface with Google Earth has been developed (see Figure 20).

## 4. Main Conclusions and Perspectives

### 4.1. Main Conclusions

At present, the ballast layer of the general speed railway line in China is mainly maintained based on the gross passing load. This easily leads to over-repair or under-repair for the ballast layer. This is contrary to the concept “state repair” for railway engineering infrastructure. In the past two decades, European countries and the United States have focused on the rapid quantitative assessment of the ballast layer and fouled ballast layer classification using GPR inspection. The combined ballast layer inspection system has been developed and used on a large scale. Along with the further rapid development of railways and the scarcity of ballast resources, how to formulate ballast layer maintenance plans, rational allocation of resources, and control maintenance costs will be great challenges. The demand for rapid ballast layer condition assessment becomes more urgent. This paper focuses on the electromagnetic characteristics of the ballast layer, GPR signal processing methods, and ballast layer condition evaluation. The main conclusions and recommendations for subsequent research directions are as follows.
(1)High-frequency GPR indicators for ballast layer condition evaluation. Quantitative assessment of the ballast layer condition can be achieved by constructing radar signal characterisation indicators, calibrating them using ballast samples with different fouling levels and then establishing regression models. The GPR signal indicators include dielectric constant methods and signal time-frequency characteristics (e.g., spectral domain integration area, scan area, axis crossings, signal time-domain inflection points, wavelet signal standard deviation, air-ballast surface signal reflection amplitude standard deviation, and high frequency scattering signal envelope amplitude). The dielectric constant method to evaluate ballast fouling level has multiple explanations for one result, thus claiming detailed information on the water content, fouling material, etc. That is, to accurately identify the fouling level, information (such as the fouling material and water content) needs to be known beforehand, which limits the rapid and efficient evaluation of the ballast layer condition based only on the dielectric constant. The regression model based on the time-frequency signal needs to be matched to the ballast particle size distribution and the fouling index by sieving. The regression model should also consider the natural rainfall conditions. High-frequency signal scattering envelope amplitude can be used to obtain clean ballast thickness, but how to set the threshold for distinguishing clean and fouled ballast based on the GPR signal, remains an unsolved problem.(2)Quantitative evaluation of the ballast layer under diverse field conditions. The frequency-domain integration area, scan area, axis crossings, and three GPR indicators have the advantages of convenient and efficient calculation and strong sensitivity. Based on these GPR indicators, we can directly see the ballast layer condition by different colours. To better perform quantitative evaluation of the ballast layer, follow-up studies should consider the railway line type (general speed line, high-speed line, heavy-haul line load, passenger dedicated lines, etc.). In addition, we should also consider the characteristics of the subgrade, bridge, tunnels, and other special structures. This means that it is necessary to carry out multi-scene data collection and calibration tests in different regional environments to establish a complete quantitative evaluation standard for the ballast layer condition and to guide the formulation of ballast layer maintenance strategies.(3)Rapid inspection equipment for ballast layer condition. To adapt to the large scale of railway infrastructures and large railway network management, focus should be placed on the development of a special detection system (equipment and software) using GPR for ballast layers. The system can be flexibly mounted on rail inspection vehicles, inspection vehicles and trolleys to achieve rapid and periodic detection of ballast layers.(4)Ballast layer condition evaluation based on multiple data. The multiple data can include, for example, rail inspection, track stiffness, and ballast layer condition. These multiple time-synchronous data from track inspection can establish a comprehensive evaluation system for ballasted tracks, which can effectively guide the maintenance strategy.

### 4.2. Perspectives

Using GPR to inspect the railway ballast layer has many advantages, such as noncontact, non-destructive to the ballast layer, fast inspection, and continuous measurement of the line rather than fixed point. However, the following limitations should be studied to overcome the disadvantages of using GPR for ballast layer inspection. Analysing the GPR signal from the ballast layer is only an indirect means to reflect ballast fouling because the process is performed through the conversion of electromagnetic signal characteristics to ballast fouling by means of calibrated numerical relationships. There may be errors in certain railway line areas caused by electromagnetic interference. The railway line areas (special structures) include turnouts, bridges with rail guards, etc., where the signal cannot be correctly detected because of iron interference.

The application of GPR can be applied to the equipment currently used in railway line inspection, such as rail inspection vehicles, so that installation costs are not very high. In addition, the use of GPR inspection on the ballast layer enables the optimisation of regular preventive maintenance, which can be operated based on the fouling degree of the ballast layer. Then, the rational allocation of the limited maintenance capacity (large machines, stuff, time, etc.), which can save significant resources and money (machinery, manpower, etc.) for railway maintenance.

## Figures and Tables

**Figure 1 sensors-22-02450-f001:**
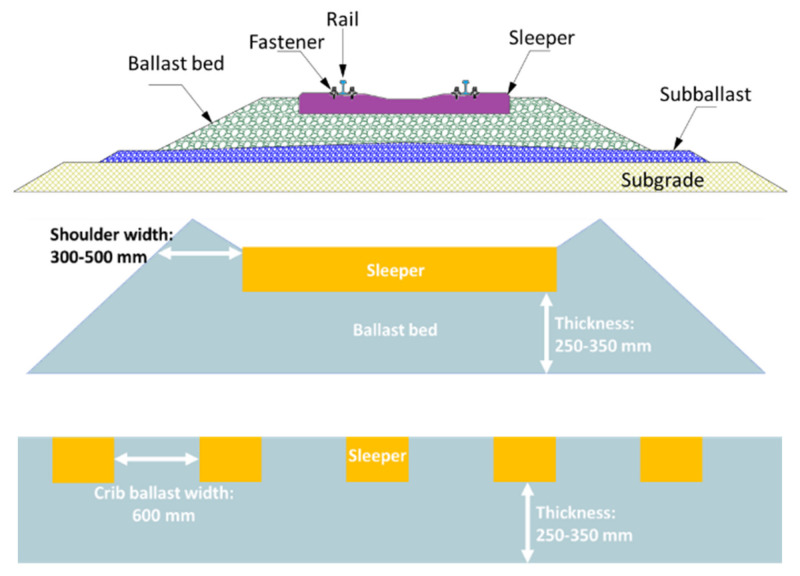
Ballasted track (figure reproduced from [22]).

**Figure 2 sensors-22-02450-f002:**
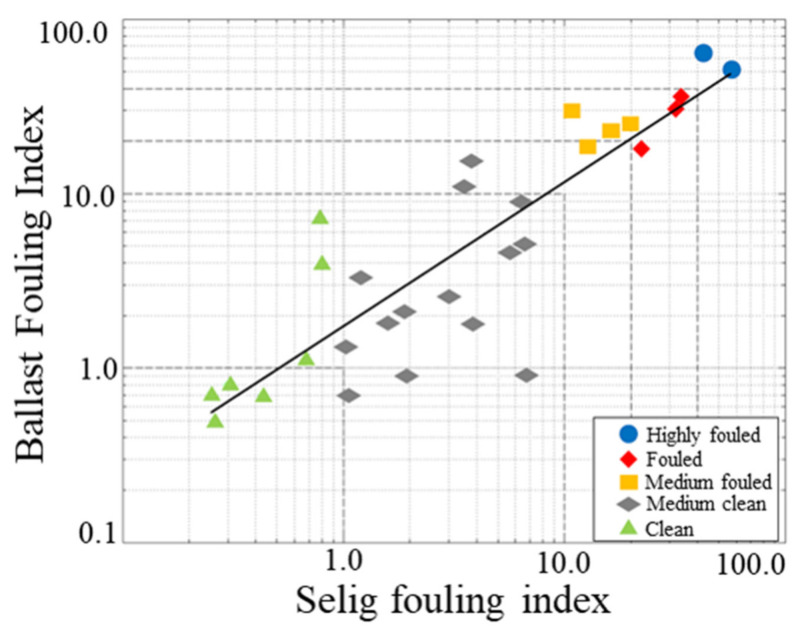
Relationship between sample Selig fouling index and BFI values.

**Figure 3 sensors-22-02450-f003:**
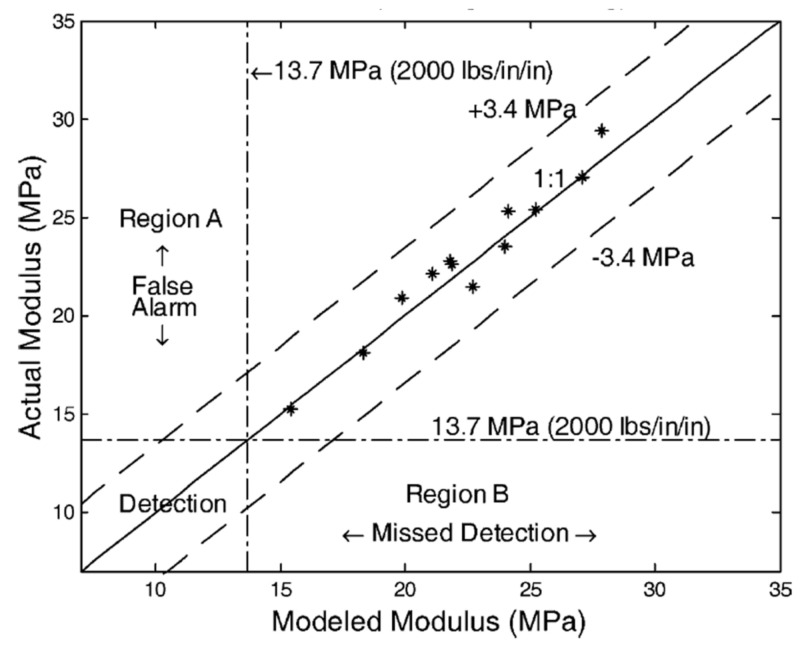
Comparison of estimated and measured rack modulus (figure reproduced from [94]).

**Figure 4 sensors-22-02450-f004:**
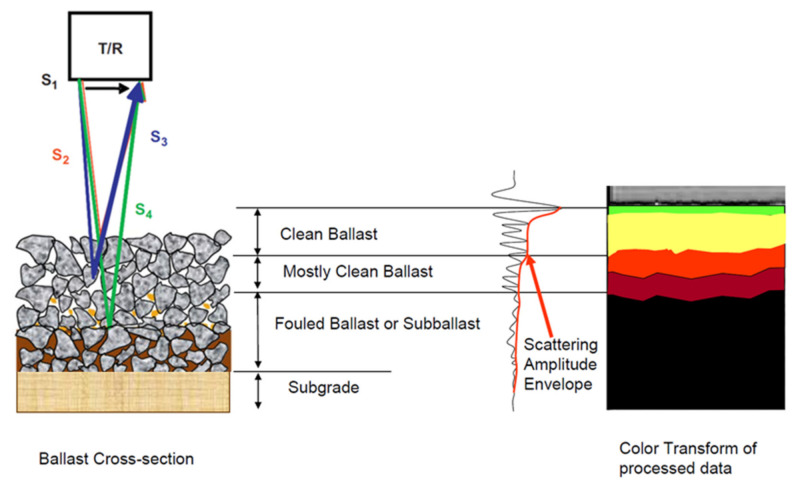
Principle of GPR detection technology for ballast layer (figure modified after [23]).

**Figure 5 sensors-22-02450-f005:**
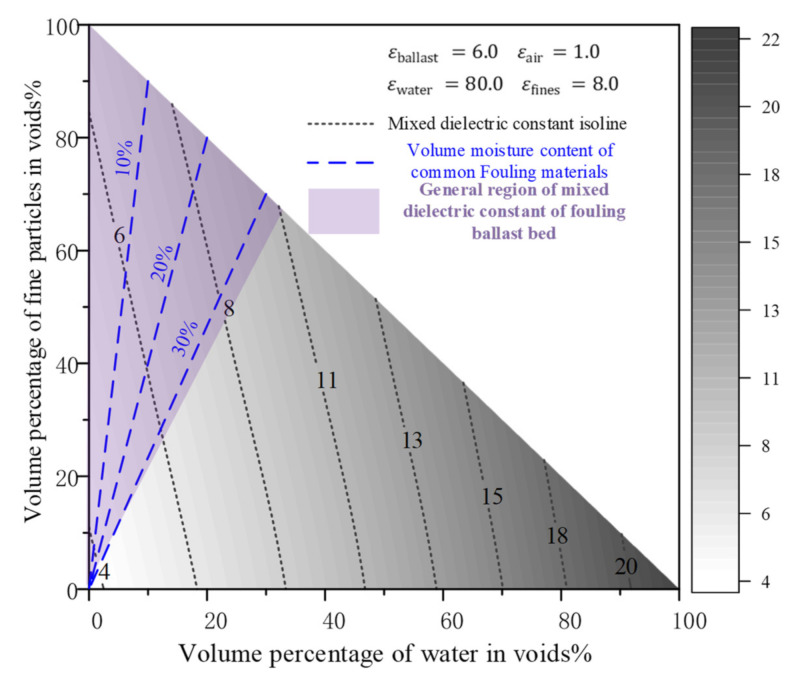
Distribution of dielectric constants for ballast layers at different fouling levels.

**Figure 6 sensors-22-02450-f006:**
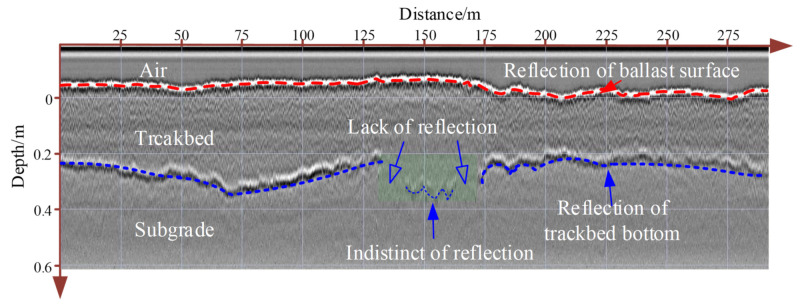
GPR signal reflection from ballast layer [89].

**Figure 7 sensors-22-02450-f007:**
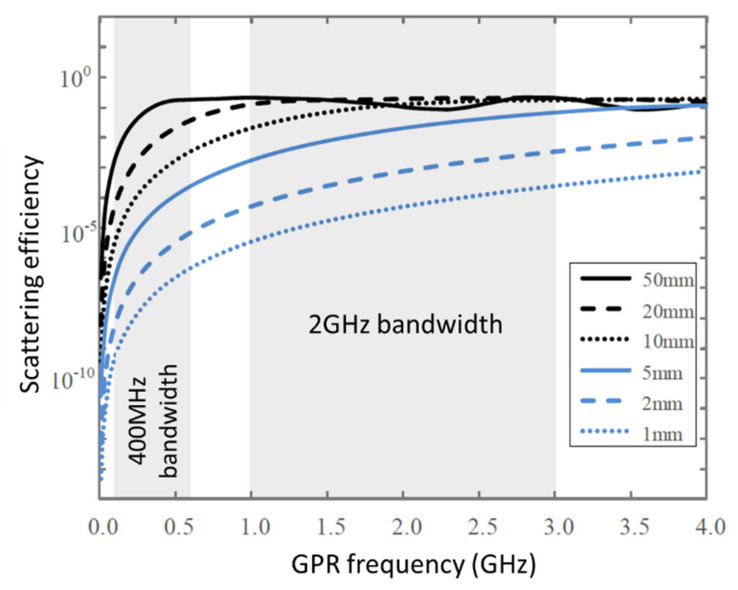
Scattering efficiency for different sizes of voids in ballast layer [86].

**Figure 8 sensors-22-02450-f008:**
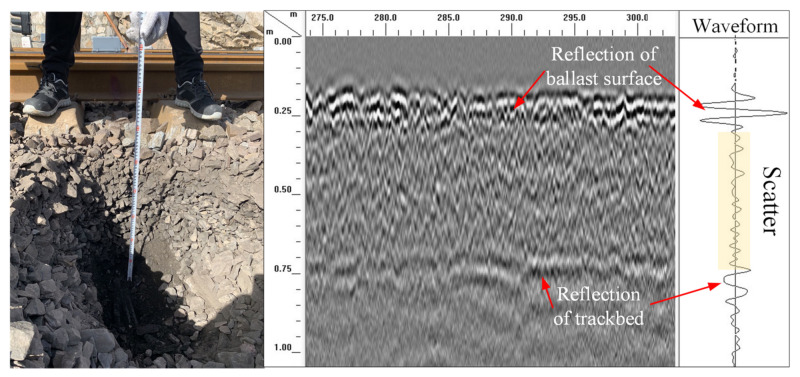
GHz ballast layer GPR signal image.

**Figure 9 sensors-22-02450-f009:**
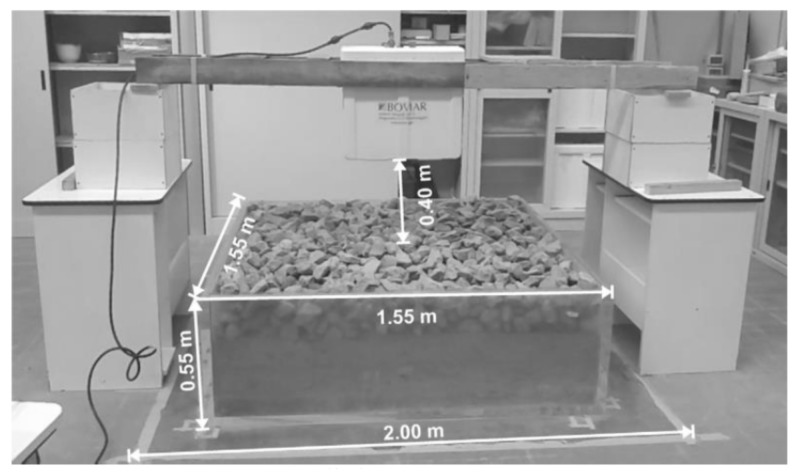
Dielectric constant measurement for ballast samples in ballast box (reproduced from [105]).

**Figure 10 sensors-22-02450-f010:**
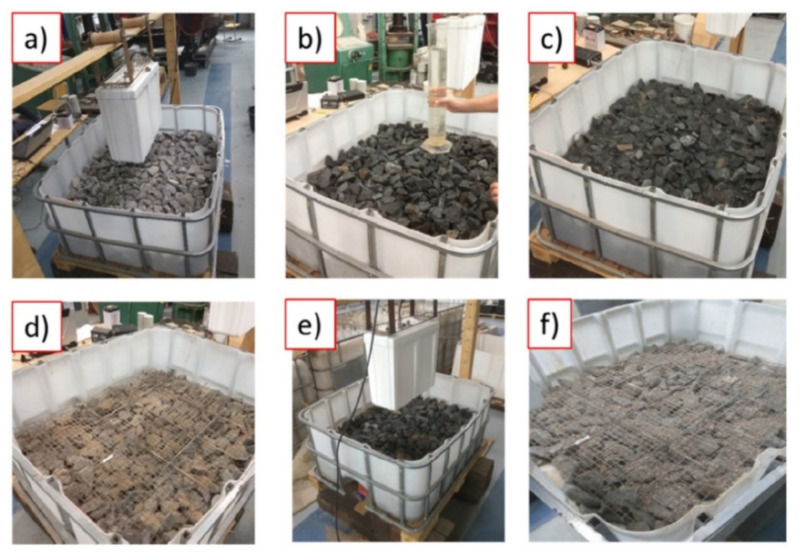
(**a**–**f**) demonstrate the whole testing procedure, including test set-up, test conditions and GPR positions. Test configurations of ballast box with ballast particles and fouling (figure reproduced from [63]).

**Figure 11 sensors-22-02450-f011:**
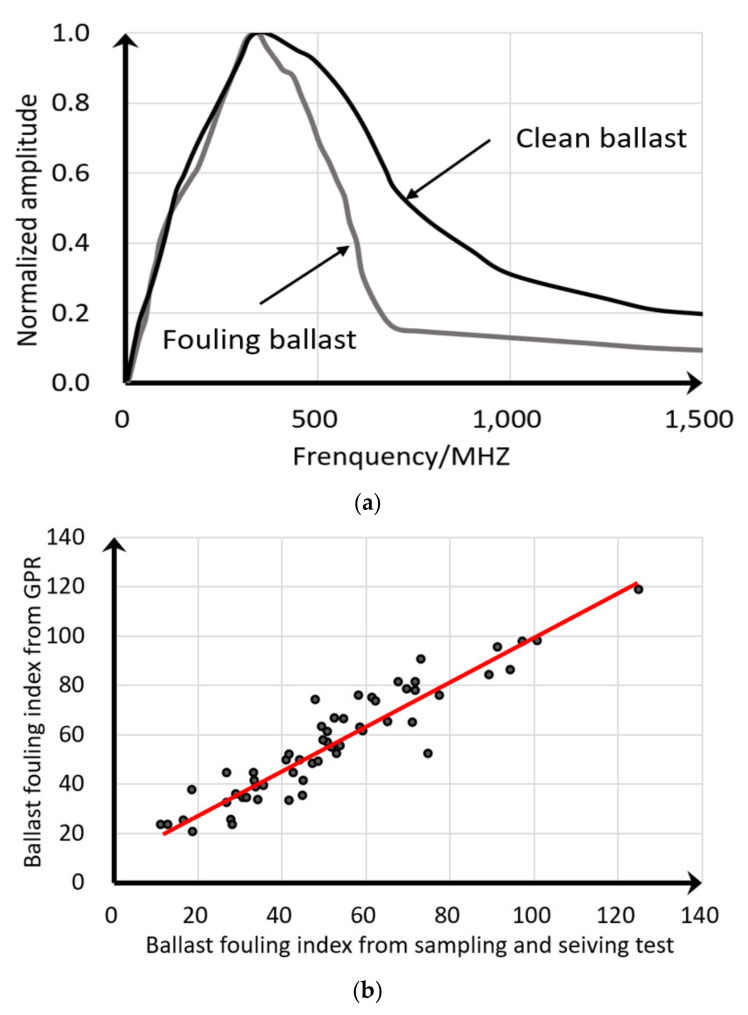
Frequency-domain GPR signal for ballast fouling level estimation (figure modified after [108]): (**a**) Frequency characteristics at different ballast fouling levels. (**b**) Ballast fouling index Correlation of GPR inspection and in situ sampling and sieving.

**Figure 12 sensors-22-02450-f012:**
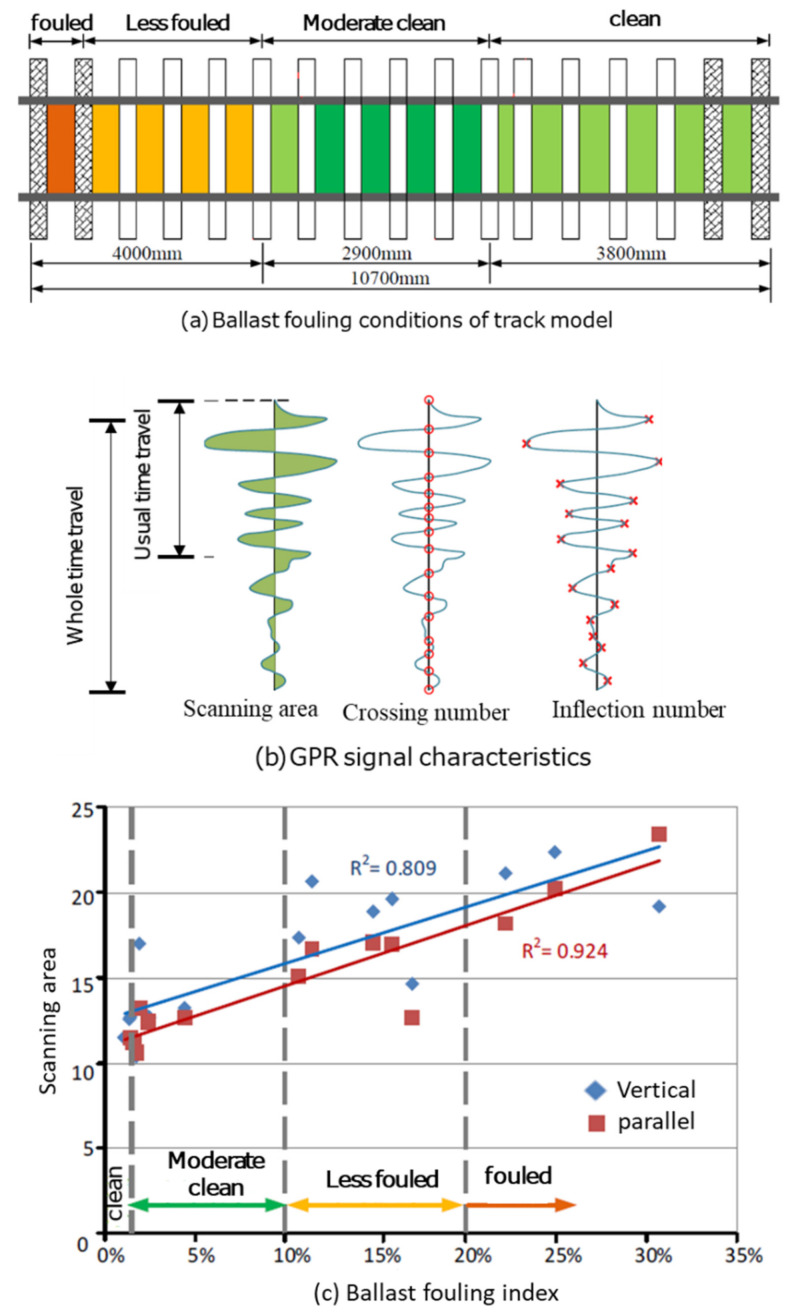
Full-scale ballast layer model and results from GPR (figure reproduced from [109]).

**Figure 13 sensors-22-02450-f013:**
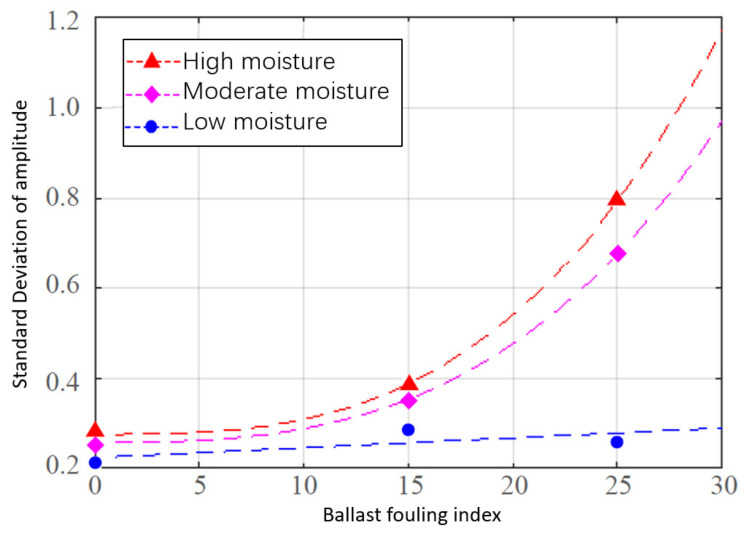
Relationship between the GPR fouling index and air-ballast reflection amplitude [112].

**Figure 14 sensors-22-02450-f014:**
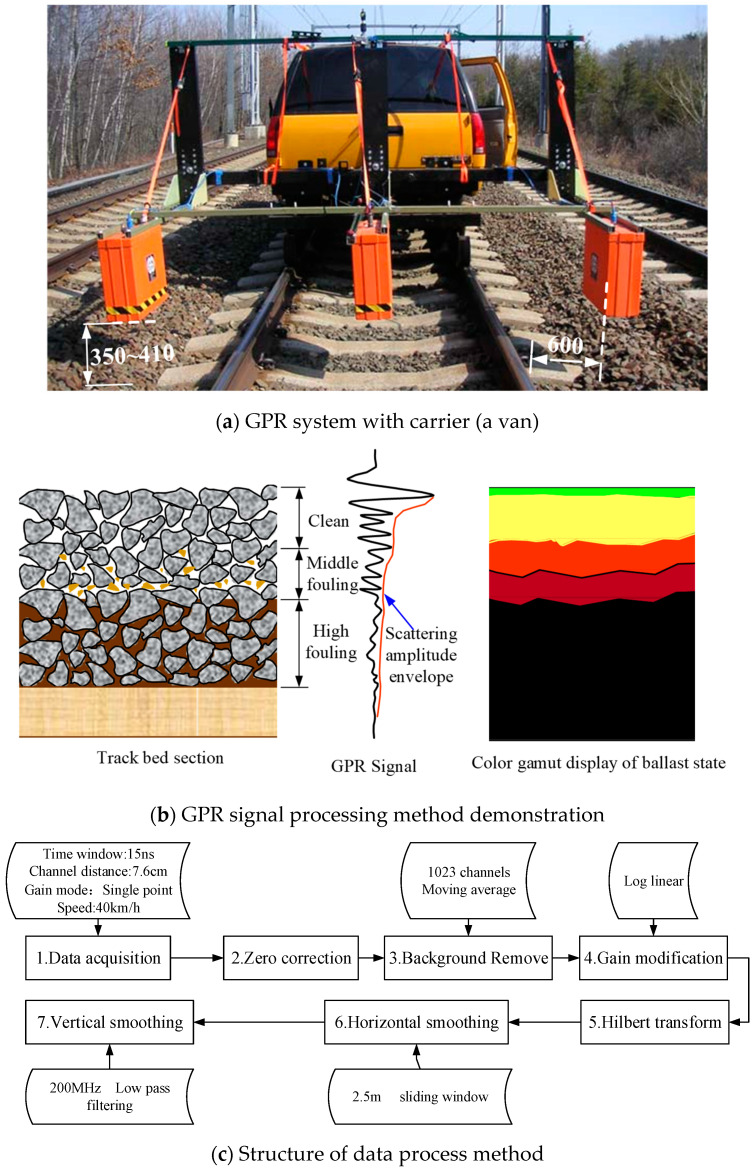
(**a**–**c**) demonstrate the GPR system and GPR signal processing method. US Federal Railroad Administration high-frequency GPR inspection (modified after [23]).

**Figure 15 sensors-22-02450-f015:**
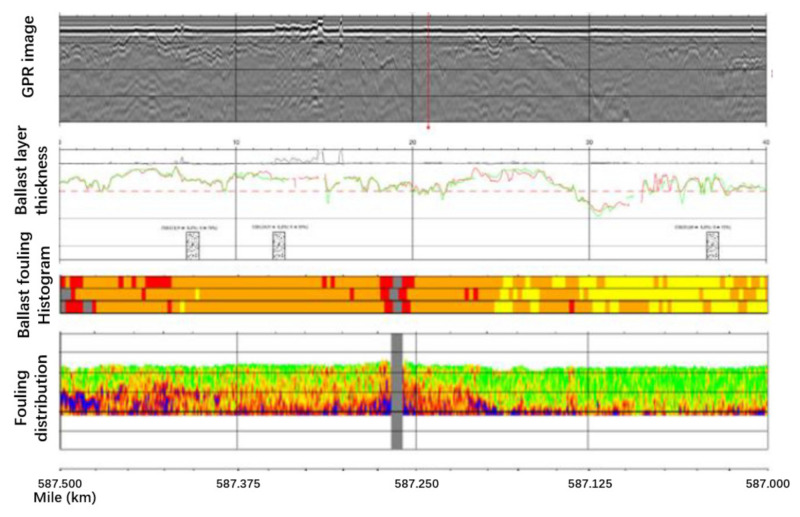
Output of Zetica’s system (figure modified after [87]).

**Figure 16 sensors-22-02450-f016:**
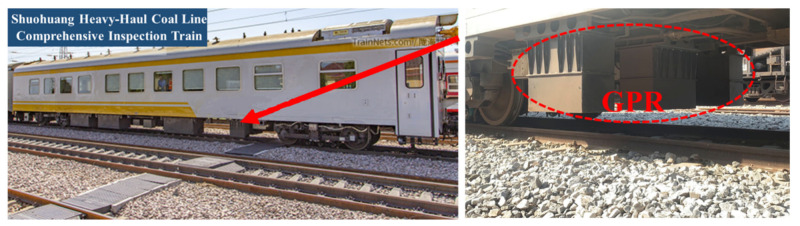
Integrated inspection vehicle with GPR system.

**Figure 17 sensors-22-02450-f017:**
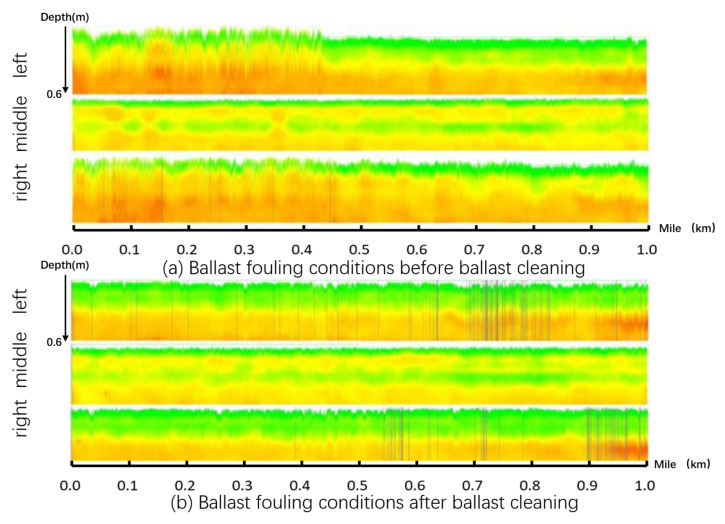
GPR fouling level demonstration of the same railway line section before and after ballast cleaning (figure modified after [122]).

**Figure 18 sensors-22-02450-f018:**
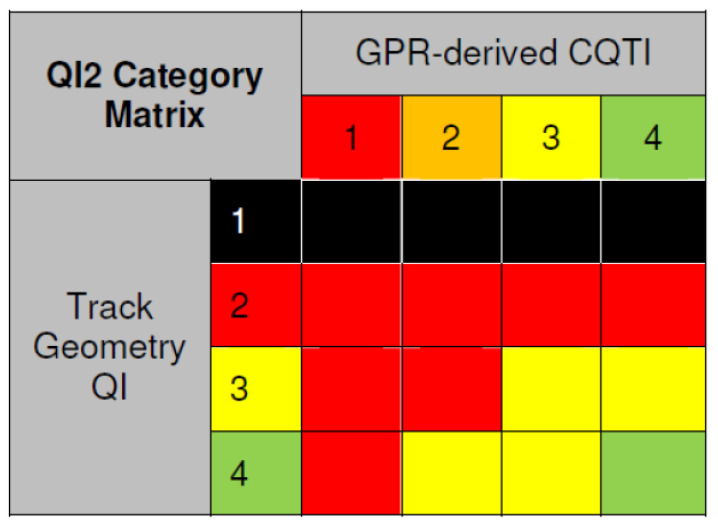
QI2 rules matrix between CQTI and track geometry QI (figure reproduced from [87]).

**Figure 19 sensors-22-02450-f019:**
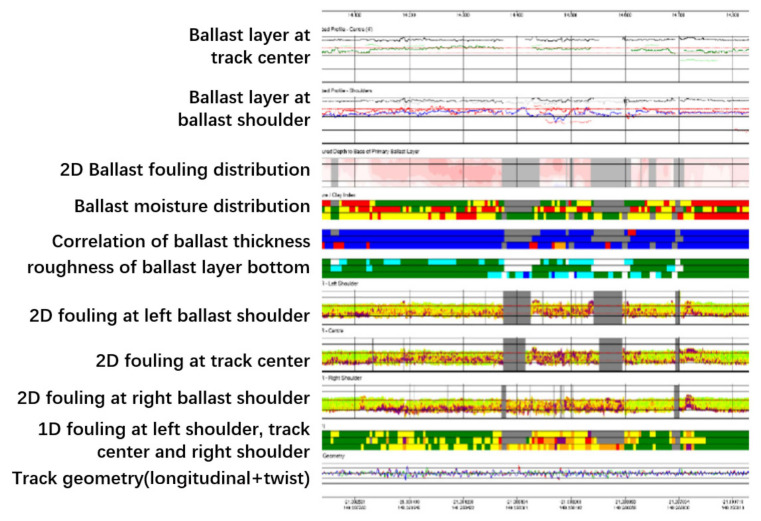
Integration of track and ballast layer indicators (figure reproduced from [87]).

**Figure 20 sensors-22-02450-f020:**
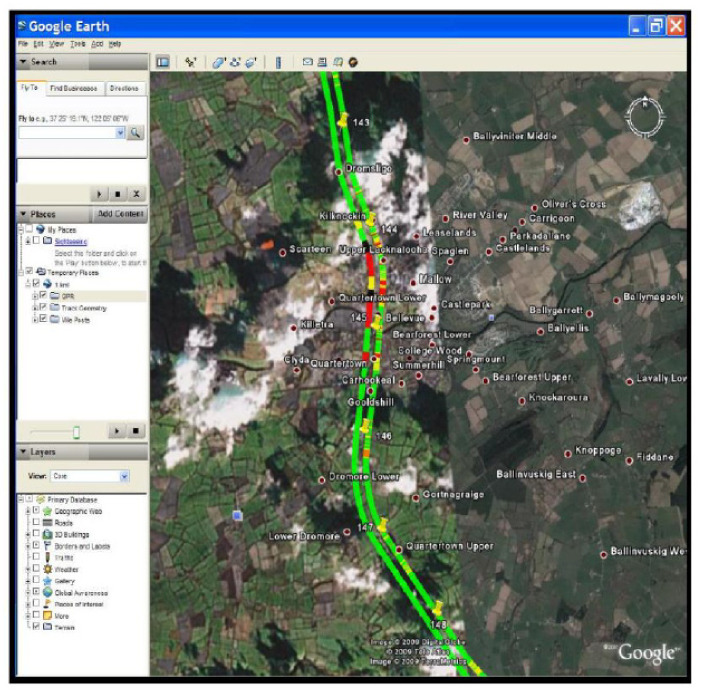
CTQI and Google Earth integration demonstration (figure reproduced from [88]).

**Table 1 sensors-22-02450-t001:** Dielectric constants of common materials in ballast layer.

Ballast Layer	Materials	Dielectric Constant
Air	Air	≈1
Ballast	Granite	5–7
	Limestone	4–8
Fouling composition	Water	80 + (related to the wave frequency)
	Sands	3–6
	Soil	2–19
	Coal	3–4

**Table 2 sensors-22-02450-t002:** Track inspection equipment system configuration, function, and recommended loading platform.

Equipment	Function	Maximum Operation Speed and Loading Platform
GPR system	Ballast layer thickness Ballast–subgrade interface Subgrade defects Mud-pumping Ballast fouling level Clean-fouled ballast interface Water content	160 km/h Railway comprehensive inspection trains, rail inspection vehicle, trolley
Track video (linear CCD camera)	Mud-pumping at surface Sleeper condition	160 km/h Railway comprehensive inspection trains, rail inspection vehicle
3D laser scanning	Ballast layer profile Surroundings Drainage	8 km/h Rail inspection vehicle, trolley

**Table 3 sensors-22-02450-t003:** Ballast fouling level standard.

Rank	Fouling Level	Fouling Index Sieving (%)	GPR Fouling Index	Maintenance Decision
1	Clean	<10	<2.0	No need for cleaning or renewal
2	Medium clean,	10~20	2.0~3.5	Keep focusing
3	Medium fouled,	20~25	3.5~4.1	On the plan list
4	Fouled	25~30	4.2~4.8	High level on the plan list
5	Highly fouled	≥30	≥4.8	Prior to others

## Data Availability

Not applicable.

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
