# Peer review of "State-of-the-Art Review of Ground Penetrating Radar (GPR) Applications for Railway Ballast Inspection"

_sensors, 2022, doi:10.3390/s22072450_

Round 1

Reviewer 1 Report

This manuscript offers a review of the GPR method applied to railroad track inspection and Non-destructive testing. Overall the manuscript provides both details on specific processing and GPR equipment applied to ballast condition inspection. The organization of the manuscript could be improved in addition to shortening the text by combining section 4 and the introduction. The GPR theory section should be improved and should provide references to a textbook or earlier publications that provide the theoretical aspects of the method. The scope of this manuscript is broad enough that the authors should include multi-offset GPR data acquisition techniques as a potential future benefit - see for example WARR machine from Sensors and Software. There are small-scale improvements to be made from reviewing the words chosen or syntax of the sentences, however, the objective of the manuscript is evident in its present form. I have broken my comments into general / big comments that could affect multiple sections and specific comments directed at a particular line or figure.

General comments:
1) Introduction seems to be written as overall review for Ballast problems and then solutions. Then starting at line 98, the tone shifts to lump all sources of problems and solutions into a few lines and pivot to assessment of the ballast. The assessment of the ballast is clearly where this contribution needs to be directed. And because of the title of the article, it seems that if you incorporated material from later in the manuscript, you could avoid this sudden tone shift. Separating the introduction into Problems with Ballast, Problems caused by Ballast - Track geometry, Previous solutions, and current examples of GPR applied to railbeds. Then the manuscript can review the state-of-the-art efforts of researchers to improve how society sees the GPR data. 

2) For a review paper in using GPR on ballast assessment, where the moisture content is important, the authors should include a discussion on multi-offset GPR data collection (i.e. WARR type measurements) and indicate the research being completed to rapidly monitor water content for agriculture. For example see van Overmeeren et al 1997., or Lambot et al 2006 or Moghadas et al 2014 (Estimation of the near surface soil water content during evaporation using air-launched ground-penetrating radar).

3) In the background GPR information you present, you should mention or show how the velocity of a GPR signal is determined from the acquired data (two-way traveltime at a known transmitter-receiver offset) or provide a citation for the audience to locate more details on the method.  This can then tie into the discussion of the quantitative GPR section where the Ballast Thickness is known and thus dielectric constant can be determined in a straight-forward manner.

4) The arrival of a signal is the first deviation in the waveform, for example the red dashed line in your Fig. 4 is too late by almost 3/4 of the GPR signal wavelet. In addition, the arrivals (and thus red line) should parallel the arrival times, hence should dip more significantly in this figure between 175 and 185 m, for example. 

5) It seems from the text that all of your figure captions should indicate the source of the figure? I.e. Figs 3,4,5,6,9,11 and possibly others. 

Specific Comments:

Line 30. Would be helpful to refer to Figure 1 here.

Line 36. You provide the results of ballast fouling, however, the reasons that lead to the deterioration of the ballast should also be provided here. You provide this at line 55, but it would make more sense to talk about cause before effect.

Line 60. You indicate that 20% is second largest percentage, but in previous line you indicate 70%. Please revisit and make this more clear.

Line 62. Incorrect word usage - you say resource but source is the word that you mean / should use.

Lines 55 to 84. Your text seems to suggest that you are providing the percentage of each of the 3 types of fouling sources that occur, however, your percentages do not add up to 100% even though source 3 is stated as being "not much", which I interpret to be less than 5%. I assume that the percentages that you are providing are coming from the literature cited, but you should rephrase these sections to make it clear that you aren't trying to generalize that source 1 (mechanical breakage) is somewhere between 20 and 70% of fouling for a given balast, source 2 (infiltration) is 58%, and then source 3 is between 22% (100 - 58 -20) and negative 28% (100 -58 -70). 

Line 88. What do you mean by "hand machine can such the coal"? Please revisit and rephrase.

Line 92. Resource should be source.

Line 95. Please rephrase - "...grows more until to the ballast layer surface."

Line 101. The sources (reasons) for ballast fouling was just discussed above this point, so are you saying that you didn't exhaust all sources in your review?

Line 105. Rather than "decides", you mean "decisions". 

Line 168. The GPR signal velocity in air is the speed of light in air, its not approximate. Do you mean that it is approximately the speed of light in a vacuum or free-space? And this section is written in a confusing way, you seem to suggest that the EM velocity dictates dielectric constant. The material properties dictate the EM velocity and thus it would be better to write that the dielectric constant of air is approximately 1 which results in an EM wave velocity of approximately 0.299792 m/ns. 

Line 212. Conductivity has negligible effect on EM velocity when considering dry lithic material. As conductivity values increase, for example salt water or metal, the EM wave velocity is significantly affected.

Line 214. Conductivity is only dependent on EM wave frequency for some material, like water.

Lines 212 to 216. The theoretical development here should be grounded and cited to a relevant reference. For example, Chapter 1 in your citation [80 - Jol] which is by Annan, P. There are several other publications that provide the relevant background on EM wave propagations and common assumptions used in different applications, in case you prefer to expand your reference section and cite those publications. 

Lines 233 and 234. Fig. 4 doesn't show any information regarding fouling along the profile distance. The arrival time of the signal changes, which may be just simply caused by a change in elevation of the radar antenna relative to the ballast.

Line 292. "standard PSD" but PSD is not defined. This should be defined clearly, particularly since it is used several times (Lines 417 and 418).

Fig. 4. The shortening of Reflection to Reflex is not appropriate as Reflex is a software package. I suggest changing to "refl" and then specify in the caption that refl is short for reflection.

Line 362. 500 M - seems to missing a portion of the abbreviation. 

Line 377. How are specific wavelets coupled? Please revisit and rephrase to make this evaluation process more clear. I believe you mean compared, but how are they compared - cross-correlation or subtraction to determine the deviation?

Fig. 11. Legend isn't complete.

Line 430. Please include more detail on the air-coupled antennas. Was there one Transmitter and 2 receivers, or 3 transmitters and 3 receivers? 

Line 433. "upon" I believe you mean above.

Fig. 13. It doesn't seem to fit in this location, the comparison between fouling indexes seems like it should be in background information and not presented in the GPR specific sections. 

Fig. 14. This graphic is difficult to see text and details. Please improve the resolution or determine which aspects of this figure are important and hide the portions that aren't important and can't be observed by the reader.

Line 488. The ballast moisture will alter the two-way traveltime observed in the GPR data, so if the velocity (or dielectric constant) is not known, then the depth will be calculated incorrectly. Thus I disagree with your assertion here, specifically that because the thickness of the ballast doesn't change due water content, the GPR reflection will still work for this method. I suggest rephrasing.

Section 3.4. GPR application section. This section seems to summarize the use of GPR on railways or the data collection portion of the application of GPR to railways. Moving this section earlier in the manuscript may provide some of the audience motivation for working to realize that your summary of the ballast analysis methods are meant to help identify the portions of the GPR signal that indicate the ballast needs to be cleaned as in Fig. 16a vs the portions that don't Fig. 16b.

Section 4. This section does not present any use of GPR. It seems that this section should be used as motivation for why ballast fouling needs to be understood since there is correlation between fouling and track geometry. 

Section 5. This section illustrates how GPR fits into a full railbed evaluation. Similar to Section 4, it seems more like it should be summary section than a separate section. 

Line 635. "The dielectric constant method to evaluate ballast fouling level has multiple explanations for one result." This sentence is either confusing or should not occur in the summary. This seems to suggest that the method is not understandable or doesn't produce a singular explanation. If this is what you want to say, then you need to be sure to show the evidence for this conclusion earlier in the manuscript and then refer the reader to it here in the summary. 

Author Response

Please see the attachement. we have revised our paper accordingly, and they are quite good advices.

Reviewer 2 Report

  • revisit the abstract,
  • update the rereferences by considering 2020-2022,
  • it would be if you add clearly the advantages and limitations of GPR in general and wrt the current application,
  • with regard to the current state of the art, what would be the cost of the application of this technique? 

Author Response

(The authors gave the same response as above.)

Reviewer 3 Report

Really, it is an interesting and useful investigation.

However, some revisions are necessary (see below).

(1) In the Introduction, some additional references to similar investigations must be included, for example: 

Alperovich, L., Eppelbaum, L., Zheludev, V., Dumoulin, J., Soldovieri, F., Proto, M., Bavusi, M. and Loperte, A., 2013. A new combined wavelet methodology applied to GPR and ERT data in the Montagnole experiment (French Alps). Journal of Geophysics and Engineering, 10, No. 2, 025017, 1-17.

(2) Table 2 must be extended.

(3) Figure 4: Distance and Depth must be given in the same physical dimension (meters).

(4) Figure 9A is of low quality.

(5) Figure 16. Where are the scales?

(6) Figure 18 needs in detailed explanation.

(7) Section 6 must be renamed. Maybe, "Main Conclusions"? 

Author Response

Thank you very much for your help with our paper. we have revised our paper accordingly. Please check the attached document.

Round 2

Reviewer 1 Report

The manuscript revision has improved the contribution significantly. I look forward to seeing the published work. 

The only concerns I have with your manuscript are errors in Figure numbers, figure references in text (appearing as Error code), equation numbers and references, and table references in text.